# Differential Myocardial Responses in Male and Female Rats with Uremic Cardiomyopathy

**DOI:** 10.3390/ijms26052259

**Published:** 2025-03-03

**Authors:** Beáta Bódi, Rebeka Rita Vágó, László Nagy, Arnold Péter Ráduly, András Gulyás, Klaudia Kupecz, Lilian Azar, Fanni Magdolna Márványkövi, Gergő Szűcs, Andrea Siska, Gábor Cserni, Imre Földesi, Zoltán Papp, Márta Sárközy

**Affiliations:** 1Division of Clinical Physiology, Faculty of Medicine, University of Debrecen, H-4032 Debrecen, Hungary; bodibea0509@gmail.com (B.B.); vgrebi@gmail.com (R.R.V.); nagy.laszlo44@gmail.com (L.N.); raduly.arnold@med.unideb.hu (A.P.R.); pappz@med.unideb.hu (Z.P.); 2Department of Cardiology, Division of Cardiology, Faculty of Medicine, University of Debrecen, H-4032 Debrecen, Hungary; 3Department of Pathophysiology, Albert Szent-Györgyi Medical School, University of Szeged, H-6720 Szeged, Hungary; ndrsglys@gmail.com (A.G.); kupeczklau@gmail.com (K.K.); lilianazar020@gmail.com (L.A.); 4Department of Biochemistry, Interdisciplinary Center of Excellence, Albert Szent-Györgyi Medical School, University of Szeged, H-6720 Szeged, Hungary; marvanykovi.fanni@gmail.com (F.M.M.); szucs.gergo@med.u-szeged.hu (G.S.); 5Department of Laboratory Medicine, Albert Szent-Györgyi Medical School, University of Szeged, H-6720 Szeged, Hungary; siska.andrea@med.u-szeged.hu (A.S.); foldesi.imre@med.u-szeged.hu (I.F.); 6Department of Pathology, Albert Szent-Györgyi Medical School, University of Szeged, H-6720 Szeged, Hungary; cserni.gabor@med.u-szeged.hu

**Keywords:** passive stiffness, active force, permeabilized cardiomyocyte, diastolic dysfunction, left ventricular hypertrophy, uremic cardiomyopathy

## Abstract

Uremic cardiomyopathy, characterized by diastolic dysfunction, left ventricular hypertrophy (LVH), and fibrosis, is a common cardiovascular complication of chronic kidney disease (CKD). Men are at a higher risk for cardiovascular and renal diseases, compared to age-matched, pre-menopausal women. We aimed to investigate the influence of sex on the severity of uremic cardiomyopathy through the characterization of functional and molecular indices of myocardial remodeling in a rat model. CKD was induced by a 5/6 nephrectomy in 9-week-old male and female Wistar rats. Serum and urine tests, transthoracic echocardiography, left ventricular (LV) histology, and quantitative reverse transcription polymerase chain reaction (RT-qPCR) were performed at week 8 or 9. Moreover, LV alterations were also tested in permeabilized cardiomyocytes (CMs) by force measurements and Western immunoblotting. CKD resulted in the development of a more severe uremic cardiomyopathy in male rats—including LVH, LV diastolic dysfunction, and fibrosis—than in female rats, where only LVH was observed. A uremic cardiomyopathy was also associated with a decrease in maximal Ca^2+^-activated force (F_max_) in CMs of male rats. Additionally, increases in CM Ca^2+^-independent passive stiffness (F_passive_) and decreases in cardiac myosin-binding protein C (cMyBP-C) phosphorylation levels were significantly larger in male than female rats. In conclusion, a uremic cardiomyopathy involved cardiac remodeling in both sexes. Nevertheless, male rats exhibited more pronounced signs of macroscopic and microscopic alterations than their female counterparts, illustrating a sex-dependent component of uremic cardiomyopathy.

## 1. Introduction

Chronic kidney disease (CKD) is one of the most rapidly growing non-communicable diseases, affecting 7–12% of the global population, largely due to the increasing prevalence of common primary causes such as aging, diabetes mellitus, and hypertension [1,2]. Interestingly, the age-standardized global prevalence of early-stage CKD (G1–G3, GFR > 30 mL/min/1.73 m^2^) is higher in women compared to men [3]. In contrast, men are characterized by higher all-cause mortality rates at all stages of pre-dialysis CKD. Mortality rates of end-stage renal disease (ESRD) patients are similar for both sexes [3]. CKD, irrespective of its severity, predisposes to a five to tenfold higher risk of cardiovascular diseases (CVDs) and premature death due to CVDs than in the age-matched non-CKD population [4].

CKD-associated chronic structural, functional, and electrophysiological remodeling of the heart is termed uremic cardiomyopathy [5]. It is characterized by left ventricular hypertrophy (LVH), interstitial fibrosis, diastolic and systolic dysfunctions, capillary rarefaction, and an increased susceptibility to further injuries, such as arrhythmias and acute myocardial infarction [5]. The prevalence of LVH increases with the CKD’s progression [6,7]. Moreover, the severity and persistence of a LVH are strongly associated with cardiovascular events and mortality risk in CKD patients [8]. Furthermore, uremic cardiomyopathy is recognized as the primary cause of death in ESRD patients [9]. Nonetheless, to date, hypothetical sex-dependent characteristics in uremic cardiomyopathy, as suggested by epidemiological studies, have not been investigated in detail.

Several factors contribute to the development of a uremic cardiomyopathy, though the precise molecular mechanisms remain unclear. Putative disease mediators include CKD-specific factors, like circulating uremic toxins and renal anemia, and CKD-unspecific factors, such as hemodynamic overload, over-activation of the sympathetic nervous system and renin–angiotensin–aldosterone system (RAAS), hypertension, endothelial dysfunction, inflammation, and increased nitro-oxidative stress [10,11]. Although a previous study implicated that men and women may be equally susceptible to developing a LV hypertrophy and fibrosis in uremic cardiomyopathy [12], structural and functional characteristics of myocardial remodeling and their potential differences between the two sexes have not been extensively studied.

In this study, we focused on sex-specific differences in myocardial remodeling in a rat model of uremic cardiomyopathy induced by a subtotal (5/6) nephrectomy. We examined the development of LVH, myocardial fibrosis, LV CM function, and changes in myofilament protein phosphorylations in both sexes (Figure 1). Understanding the molecular mechanisms of the uremic cardiomyopathy can pave the way for sex-specific therapeutic strategies for CKD-associated CVDs.

## 2. Results

### 2.1. Males and Females Developed CKD of Similar Severity

To confirm the development of a CKD induced by the 5/6 nephrectomy, urine and serum parameters were analyzed at weeks 8 and 9, respectively (Figure 1). The serum urea and creatinine levels were comparable in sham-operated males and females (Figure 2A,B). Both male and female 5/6 nephrectomized rats exhibited significantly elevated serum urea (12.68 ± 0.83 vs. 6.86 ± 0.37 mmol/L in males and 12.19 ± 0.69 vs. 6.5 ± 0.52 mmol/L in females, respectively, *p* < 0.05) and creatinine levels (69.25 ± 6.2 vs. 26.5 ± 1.21 μmol/L in males and 63.08 ± 4.25 vs. 31.8 ± 1.99 μmol/L in females, respectively, *p* < 0.05) compared to the sex-matched sham-operated animals, indicating a decline in kidney function in both sexes (Figure 2A,B).

There were no significant differences in serum carbamide and creatinine levels between the male and female 5/6 nephrectomized rats (12.68 ± 0.83 vs. 12.19 ± 0.69 mmol/L and 26.5 ± 1.21 vs. 31.8 ± 1.99 μmol/L, respectively). Creatinine clearance was significantly reduced in both male and female 5/6 nephrectomized rats, compared to their sex-matched sham-operated counterparts (1.59 ± 0.22 vs. 3.05 ± 0.22 mL/min/day in males and 1.03 ± 0.08 vs. 1.69 ± 0.12 mL/min/day in females, respectively, *p* < 0.05), suggesting similar levels of impaired kidney function in both sexes (Figure 2C). Sham-operated females had significantly lower creatinine clearance (1.69 ± 0.12 mL/min/day) than sham-operated males (3.05 ± 0.22 mL/min/day) (Figure 2C). Urine creatinine concentrations were comparable between the sham-operated animals (4947 ± 848 mmol/L in males and 5101 ± 637 mmol/L in females, respectively) (Figure 2D). It was not significantly different between the male CKD and sham-operated groups (3678 ± 519 vs. 4947 ± 848 mmol/L in males, respectively, *p* = 0.323); however, it tended to decrease in the female 5/6 nephrectomized rats, compared to sex-matched sham-operated animals (2938 ± 265 vs. 5101 ± 637 mmol/L, respectively, *p* = 0.081). Urine volume showed no significant differences between the sham-operated male (21.78 ± 2.6 mL/day) and female groups (18.22 ± 2.22 mL/day). Still, it was significantly higher in both male and female 5/6 nephrectomized rats compared to the sex-matched sham-operated animals (35.33 ± 2.75 vs. 21.78 ± 2.6 mL/day in males and 32.67 ± 2.81 vs. 18.22 ± 2.22 mL/day in females, respectively, *p* < 0.05, respectively), indicating the onset of the polyuric phase of CKD (Figure 2E). Urine protein concentration was significantly lower in sham-operated females (39.61 ± 5.83 mg/dL) compared to males (87.29 ± 14.23 mg/dL) (Figure 2F). However, urine protein concentration was significantly increased in both male and female 5/6 nephrectomized rats compared to the sex-matched controls (481 ± 73.6 vs. 87.29 ± 14.23 mg/dL in males and 284 ± 51.4 vs. 39.61 ± 5.83 mg/dL in females, respectively, *p* < 0.05), reflecting a worsening of glomerular function in CKD (Figure 2F).

### 2.2. CKD Males Exhibited More Severe Echocardiographic Signs of LVH than Females

Transthoracic echocardiography performed at week 8 assessed the impact of the CKD and sex on myocardial morphology and function (Figure 3A–G and Table 1). No significant differences were observed in LV systolic anterior and septal wall thicknesses both in systole and diastole (AWTs, AWTd, SWTs, and SWTd, respectively) between sham-operated males and females (Figure 3A–E, Table 1). However, in sham-operated females, LV end-diastolic and end-systolic diameters (LVEDD and LVESD) and volumes (LVEDV and LVESV), stroke volume (SV), and cardiac output (CO) were significantly reduced compared to sham-operated males (Table 1), likely due to the smaller heart size of females. In response to CKD, males showed significant increases in AWTs, AWTd, SWTs, SWTd, and diastolic posterior wall thickness (PWTd) compared to the sex-matched sham-operated animals (Figure 3B–D, Table 1). In females, only the AWTs were significantly thicker in response to CKD compared to the sham-operated group (Figure 3B and Table 1). Notably, in the CKD groups, AWTd and SWTd were significantly smaller in the females than in males (Figure 3C,E, Table 1). Moreover, CKD males exhibited significantly decreased left ventricular LVESV, LVEDV, SV, and CO compared to the sham-operated animals, indicating more severe concentric LVH in CKD males (Table 1). Heart rate (HR) increased significantly in the CKD females compared to their sex-matched sham-operated counterparts (Table 1).

The main systolic function parameter, ejection fraction (EF), remained unchanged between the groups (Figure 3F and Table 1). However, fractional shortening (FS), representing the percentage change in the LV diameter during systole, was significantly higher in the CKD males compared to the sham-operated males due to concentric LVH (Table 1). Notably, females had significantly higher FS regardless of CKD, attributed to their smaller LV diameters compared to males (Table 1).

Diastolic function was evaluated by measuring E- and E’-velocities, their ratio, and isovolumic relaxation time (IVRT) (Table 1, Figure 3G). The E/E’ ratio and IVRT were significantly increased in CKD males only, indicating more severe diastolic dysfunction in males compared to females with CKD (Figure 3G and Table 1).

Despite the well-known association of hypertension with CKD, and its role as an independent risk factor for LVH development, our study found no significant differences in systolic (SBP), diastolic (DBS), or mean arterial blood (MBP) pressure between the groups at week 9, regardless of CKD or sex (Table 2).

### 2.3. Both Males and Females Exhibited Ex Vivo Morphological Signs of Hypertrophy in CKD

At week 9, morphological parameters such as body weight, tibia length, heart weight, lung weight, and left kidney weight were assessed to explore the effects of CKD on ex vivo parameters in both sexes. Females had significantly lower body weight and tibia length compared to males, both in sham-operated and CKD conditions (Table 2). CKD did not affect body weight or tibia length in either sex when compared to the sham-operated animals (Table 2). Regardless of CKD, females had significantly smaller heart, lung, and kidney weights compared to males, attributable to their smaller body size (Table 2). In response to CKD, heart weight increased significantly in both males and females compared to their sex-matched sham-operated counterparts (Table 1). Notably, lung weight increased significantly in CKD males but not in CKD females compared to the sex-matched, sham-operated animals, suggesting mild pulmonary congestion and potentially a more severe uremic cardiomyopathy in males with CKD. Interestingly, the weight of the remaining one-third of the left kidney in CKD animals was markedly higher than that of the entire left kidney in sham-operated animals, indicating pronounced compensatory renal hypertrophy in CKD animals, regardless of sex (Table 2).

### 2.4. CKD Caused Cardiomyocyte Hypertrophy in Both Sexes but Severe Fibrosis Only in Males

Cardiomyocyte cross-sectional areas were assessed on hematoxylin-eosin-stained histological slides to confirm the development of LVH at the cellular level, as evaluated by echocardiography and autopsy (Figure 4A). The cardiomyocyte cross-sectional areas were significantly larger in CKD rats compared to the sham-operated rats in both sexes (419 ± 20.4 vs. 297 ± 16 μm^2^ in males and 478 ± 18.5 vs. 380 ± 11 μm^2^ in females, respectively, *p* < 0.05) (Figure 4A,B), indicating LVH development at the cellular level, for both males and females.

To validate the echocardiographic signs of cardiac remodeling in CKD, LV collagen content was assessed using picrosirius red and fast green (PSFG) staining. Only CKD males exhibited a significant increase in LV collagen content, compared to sham-operated males (9.04 ± 0.32 vs. 5.28 ± 0.18%, respectively, *p* < 0.05), indicating the development of cardiac fibrosis by week 9 (Figure 4C). In females, no significant difference in LV collagen content was observed between sham-operated and CKD groups (6.11 ± 0.11 vs. 6.9 ± 0.68%, respectively), suggesting a less severe uremic cardiomyopathy in females (Figure 4C). Notably, female CKD rats had significantly lower LV collagen content than male CKD rats (6.9 ± 0.68 vs. 9.04 ± 0.32%, *p* < 0.05). To support these findings, the expression of fibrosis markers in the LV was measured by RT-qPCR. Only CKD males showed significantly increased relative expression of collagen type I alpha 1 (Col1a1) (1.46 ± 0.07 vs. 0.93 ± 0.06, respectively, *p* < 0.05) (Figure 4D) and collagen type III alpha 1 (Col3a1) (1.71 ± 0.08 vs. 1.05 ± 0.09, respectively, *p* < 0.05) (Figure 4E) compared to their sex-matched sham-operated counterparts, confirming the histological results.

### 2.5. Myofilament Function

#### 2.5.1. Reduced Ca^2+^-Activated Force Generation in Single Cardiomyocytes from CKD Male Hearts In Vitro

Ca^2+^-activated force generation was assessed in permeabilized CMs isolated from LV tissue samples. Maximal Ca^2+^-activated force (F_max_) was significantly less in CMs of CKD male rats than in sex-matched sham-operated rats (14.85 ± 1.29 kN/m^2^ vs. 20.45 ± 1.34 kN/m^2^, respectively, *p* < 0.05; Figure 5A,B). In contrast, F_max_ was similar in sham-operated and CKD female rats (18.91 ± 1.86 kN/m^2^ vs. 15.03 ± 1.22 kN/m^2^, respectively, n = 6–9 CMs per group from at least three different hearts; Figure 5A,B). The Ca^2+^-sensitivity of force production (pCa_50_) was comparable in all experimental groups: 5.95 ± 0.01 in sham-operated male rats vs. 5.99 ± 0.02 in CKD male rats, and 5.97 ± 0.03 in sham-operated female rats vs. 5.91 ± 0.02 in CKD female rats (Figure 5C,D).

#### 2.5.2. Increased Passive Stiffness in CKD Groups In Vitro

Ca^2+^-independent passive stiffness (F_passive)_ was higher in the LV of CKD male rats (1.89 ± 0.09 kN/m^2^) than in the sex-matched sham-operated rats (0.85 ± 0.06 kN/m^2^ in sham-operated female rats, *p* < 0.05; Figure 6A). The increase in F_passive_ was higher in CKD males than in CKD females (1.01 ± 0.06 kN/m^2^ vs. 0.67 ± 0.07 kN/m^2^, *p* < 0.05, n = 6–9 cardiomyocytes per group from at least three different hearts; Figure 6A). Titin is the primary intracellular factor determining the F_passive_ in the contractile machinery. Biochemical investigations revealed no differences in the levels of titin protein phosphorylations between the experimental groups (Figure 6B). Since the overall level of titin phosphorylation remained unchanged, it can be assumed that any alterations are likely site-specific rather than global.

#### 2.5.3. Hypophosphorylation of cMyBP-C Protein in CMs Affected by Uremic Cardiomyopathy

A phosphorylation analysis of cardiac troponin I (cTnI) and cardiac myosin-binding protein C (cMyBP-C). was performed using specific antibodies in Western immunoblot assays. The phosphorylation of cTnI at PKA- and PKC-specific sites Ser-22/23, Thr-144, and Ser-43 did not differ among the study groups (Figure 7A–C). In contrast, the phosphorylation of cMyBP-C at the PKA-specific Ser-282 site was significantly hypophosphorylated in both CKD male and female hearts (0.28 ± 0.03 and 0.67 ± 0.03, respectively, *p* < 0.05, n = 12–14 independent determinations) compared to the sham-operated groups (1.00 ± 0.03 in male hearts and 1.00 ± 0.01 in female hearts, respectively, *p* < 0.05; Figure 7D).

## 3. Discussion

Here, we report that although a subtotal (5/6) nephrectomy leads to similar degrees of CKD in both sexes, a 9-week-long period following kidney surgery evoked more severe uremic cardiomyopathy and LVH in male rats than in female rats. Here, we also show that collagen accumulation and an increase in CM passive stiffness were paralleled by a diastolic dysfunction in males. Moreover, our data highlight that the above changes are accompanied by additional CM alterations limiting myofilament function, including a decreased F_max_ in males and hypophosphorylation of the cMyBP-C protein in both sexes.

CKD is a significant risk factor for the development of CVDs, particularly in promoting cardiac abnormalities depending on the severity of CKD [13]. A 5/6 nephrectomy is a well-characterized model of CKD, mimicking the consequences of the progressive loss of functional nephron numbers [14,15]. Generally, the upper and lower poles of one kidney are removed in the first step, followed by the removal of the other whole kidney one week later to reduce the early postoperative mortality due to adaptation failure. The investigation of renal morphology changes was out of the scope of the present study, which focused on uremic cardiomyopathy. According to data from the literature, this model spontaneously develops hallmarks of human CKD, including uremia, fibrosis, capillary rarefaction, progressive renal function decline, accumulation of uremic toxins, and expression of TGF-β [16,17]. As reported here, the consequences of a 5/6 nephrectomy are consistent with the data from the literature and with our previous results [14,18,19,20,21]. We found characteristic changes associated with CKD in routine laboratory parameters, including higher serum carbamide, creatinine, and urine protein levels, as well as a reduced creatinine clearance, in both sexes. Males and females developed CKD of similar severities, based on serum carbamide and creatinine levels as well as creatinine clearance, 9 weeks after the 5/6 nephrectomy, as reported previously by our team [21]. In our present study, proteinuria was less severe in females than in males, which is consistent with the findings of Lemos et al., who also employed a 5/6 nephrectomy to induce CKD in Wistar rats [22].

The lack of severe hypertension and atherosclerosis in our model allows for the study of the direct effects of CKD on cardiac remodeling. However, it is important to acknowledge that this model does not fully replicate the complex clinical setting of CKD, where hypertension, diabetes, and other comorbidities often contribute to cardiac remodeling. Future studies incorporating these factors may provide additional insights into the multifactorial nature of a uremic cardiomyopathy. While traditional cardiovascular risk factors, such as hypertension, hypercholesterolemia, and diabetes mellitus, are highly prevalent in CKD patients, clinical trials aimed at mitigating their effects have mostly yielded negative results [14]. Both experimental and clinical studies have shown that CKD-specific risk factors, such as uremic toxins and renal anemia, as well as the overactivation of the renin–angiotensin–aldosterone system and the sympathetic nervous system, with increased nitro-oxidative stress and lower nitric oxide levels, can provoke the development of a uremic cardiomyopathy and increase the risk of further cardiovascular complications such as arrhythmias and acute myocardial infarction, regardless of pressure and volume overload [10,11].

Uremic cardiomyopathy is a common complication and a prognostic factor for cardiovascular mortality in CKD patients [10,11]. In our present study, characteristic morphological and functional changes in uremic cardiomyopathy were confirmed by echocardiography and histology conducted 8–9 weeks after nephrectomy. Animals of both sexes developed LVH; however, only the males exhibited interstitial fibrosis with marked diastolic dysfunction. These results are in line with those of previous studies conducted both by others and by us [21,23,24]. However, there are only a limited number of studies where the severities of LVH and/or cardiac fibrosis of males or females were quantified in uremic cardiomyopathy [23,25]. For example, Paterson et al. found no significant difference in LV collagen content between sham-operated and subtotal nephrectomized female rats 7 weeks after the intervention [23]. Hence, our present results confirm the above findings on missing cardiac fibrosis about 2 months following the development of uremia in female rats, and extend it with the recognition of cardiac fibrosis in male rats.

Both the Multi-Ethnic Study of Atherosclerosis and the Framingham Heart Study reported that LV mass and volume are significantly lower in healthy women than in men, even after adjusting for height and body surface area [26,27]. In our present study, the sex-based differences in heart size were also clearly demonstrated by heart weight, left ventricular end-systolic and end-diastolic diameters, stroke volume, and cardiac output in sham-operated animals. In women, pre-menopausal status is strongly associated with better diastolic function [26]. Moreover, it has also been demonstrated that estrogen acts as a direct vasodilator by promoting nitric oxide production, thereby improving myocardial Ca^2+^ handling and diastolic function in females [26]. In our present study, the diastolic dysfunction marker E/E’ was significantly lower in females than in males in the sham-operated group, which is in line with the aforementioned observations. Marked hypertension is not a typical feature of CKD models induced by 5/6 nephrectomy [14,28]. In line with this, our study found no significant difference in blood pressure between CKD animals of either sex. Consequently, diastolic dysfunction in our male CKD model may arise directly from LVH, cardiac fibrosis, and increased CM passive stiffness, independent of severe hypertension. Our echocardiographic, histological, and RT-qPCR findings indicate that female rats with CKD exhibit a less severe uremic cardiomyopathy phenotype than male rats at 8–9 weeks following subtotal nephrectomy. Additionally, our findings on uremic cardiomyopathy development in female rats are in line with those clinical observations where pre-menopausal women were associated with a relatively low risk of LVH and cardiac remodeling [29]. However, despite confirming a sex-dependent difference in uremic cardiomyopathy severity, our study does not provide a mechanistic explanation for the observed protection in females. While estrogen’s cardioprotective effects have been widely documented, the specific pathways underlying the relative resistance of female rats to severe uremic cardiomyopathy remain unclear, and warrant further investigation using omics techniques for discovering sex-based differences and further confirmatory mechanistic experiments.

Previous research has demonstrated that collagen accumulation and increased passive LV stiffness significantly contribute to diastolic dysfunction in uremic cardiomyopathy [23]. The increased LV passive stiffness was linked to changes in sarcomeric myofilament proteins, particularly cMyBP-C hypophosphorylation, impairing diastolic relaxation and exacerbating heart failure symptoms [30,31]. Our results on increased F_passive_ and cMyBP-C hypophosphorylation are in line with the above findings. Moreover, our present data are also consistent with those highlighting the role of increased extracellular collagen deposition in enhancing passive LV stiffness in CKD models [24,32]. Collectively, these findings underscore the significance of sarcomeric myofilament alterations and collagen deposition in therapeutic strategies for uremic cardiomyopathy. Our study extends the previous observations and implies that the increase in passive LV stiffness, leading to a diastolic dysfunction, relies on the combination of CM myofilament protein changes and extracellular collagen deposition in male rats with uremic cardiomyopathy. In contrast, the relatively small increase in F_passive_ in the absence of collagen accumulation was insufficient to evoke LV diastolic dysfunction in female rats with uremic cardiomyopathy. Furthermore, our study did not explore the potential molecular regulators, such as post-translational modifications of titin or other sarcomeric proteins, which may contribute to the observed passive stiffness changes. Future investigations focusing on these aspects could provide a more comprehensive understanding of the underlying mechanisms.

On top of the myofilament protein changes listed above, our data also revealed additional ones, namely those involving a decreased F_max_ in LV CMs of male uremic rats that might also be associated with hypophosphorylation of cMyBP-C. All sarcomeric protein alterations are consistent with the literature data, indicating that modifications in myofilament proteins and Ca^2+^ handling might be critical to the progression of diastolic dysfunction in CKD [18,32]. CM Ca^2+^ handling was not investigated here, although major changes in that cellular function are not probable in view of the unchanged EF values, when pCa_50_ was not altered either. In accordance with the unchanged pCa_50_ values, we did not find alterations in the phosphorylation levels of cTnI. Moreover, the total phosphorylation level of the giant sarcomeric protein titin was not altered here either. Nevertheless, this latter finding does not exclude antiparallel phosphorylation changes in specific sites of titin. Collectively, the molecular nature of the observed increase in F_passive_ remained unclear here.

All in all, the changes in the mechanical and biochemical characteristics in male rats outweighed those in female rats during uremic cardiomyopathy, and this underscored a sex-based difference in the pathophysiology of uremic cardiomyopathy. We acknowledge that further research is required to elucidate the molecular pathways coordinating the observed sex-dependent mechanisms and the related therapeutic strategies for uremic cardiomyopathy, and that uremic periods longer than those employed here might induce more progressed stages of uremic cardiomyopathy in both sexes [12]. Moreover, the duration of the uremic state in our study was limited to approximately two months following the onset of CKD. While this time frame allowed for the development of the characteristic uremic cardiomyopathy features, longer experimental periods may reveal more advanced stages of the disease and potentially modify the observed sex-related differences.

Taken together, our study revealed that a CKD-induced uremic cardiomyopathy evokes a CM hypertrophy in both sexes, but involves diastolic dysfunction, significant collagen accumulation, and an increased F_passive_ only in male rats, in a model of CKD evoked by a subtotal nephrectomy, at about two months following the onset of uremia. These differences were observed despite similar serum indices of CKD, suggestive of distinct signaling between the kidneys and the hearts, resulting in uremic cardiomyopathy.

## 4. Materials and Methods

This investigation conformed to the EU Directive 2010/63/EU and the National Institutes of Health Guide for the Care and Use of Laboratory Animals (NIH Publication No. 85–23, revised 1996). It was approved by the regional Animal Research Ethics Committee of Csongrád County (license number: X.1291/2024, the date of approval: 14 June 2024) and the University of Szeged in Hungary. All institutional and national guidelines for the care and use of laboratory animals were followed.

### 4.1. Animals

A total of 44 adult (9-week-old) female (200–250 g) and male Wistar rats (300–350 g) were randomized per body weight in this study. The acclimatization period was 1 week before the start of the experiments. Twenty animals (10-10 males and females) underwent a sham operation, and twenty-four animals (12-12 males and females) underwent a 5/6 nephrectomy to induce CKD. Animals were housed in pairs, in individually ventilated cages (Tecniplast Sealsafe IVC system, Buguggiate, Italy), and were maintained in a temperature-controlled room with 12 h:12 h light/dark cycles throughout the study. Standard rat chow and tap water were supplied ad libitum.

### 4.2. Experimental Setup

Experimental CKD was induced by a 5/6 nephrectomy. Animals underwent either a sham operation or a 5/6 nephrectomy in two phases. After the operations, both groups were followed up for 9 weeks. At week 8, cardiac morphology and function were assessed using transthoracic echocardiography (Figure 1). Moreover, the animals were placed in metabolic cages at week 8 for 24 h to measure urine creatinine and protein levels. At the termination of the experiment at week 9, body weight was measured, then the rats were anesthetized with sodium pentobarbital (Euthasol; Produlab Pharma b.v., Raamsdonksveer, The Netherlands, 40 mg/kg, intraperitoneally). Invasive blood pressure measurement was performed in the right femoral artery in a subgroup of animals. After the blood pressure measurement, the abdominal cavity was opened to collect blood from the aorta. Then, sodium pentobarbital was overdosed (Euthasol, 200 mg/kg, intraperitoneally; Produlab Pharma b.v., The Netherlands) to euthanize the rats. Then, the hearts, left kidneys, lungs, and tibias were isolated, and the blood was washed out from the collected samples in a calcium-free Krebs–Henseleit solution. The hearts, kidneys, and lungs were weighed, then the left and right ventricles were separated. The cross-section of the left ventricles at the ring of the papillae was cut and fixed in 4% buffered formalin for histological analysis. Other parts of the left ventricles were freshly frozen in liquid nitrogen and stored at −80 °C until further physiological and biochemical measurements, including permeabilized CM force measurements and sodium dodecyl sulfate (SDS) gel electrophoreses followed by Western immunoblotting for titin, cTnI, and cMyBP-C.

### 4.3. Subtotal (5/6) Nephrectomy Model

The sham operation and the 5/6 nephrectomy were performed in two phases, as described previously [18,21] (Figure 1). Anesthesia was induced by intraperitoneal injection of pentobarbital sodium (Euthasol; 40 mg/kg; Produlab Pharma b.v., Raamsdonksveer, The Netherlands). At the first operation, the 1/3 of the left kidneys on both poles were excised by ligation with sutures (5–0 Mersilk; Ethicon, Sommerville, NJ, USA). Accidental bleeding was alleviated by thermal cauterization. One week after the first operation, the animals were anesthetized again, and the right kidney was freed from the surrounding adipose tissue and the renal capsule. The right kidney was then gently pulled out of the incision. The adrenal gland was gently freed and placed back into the abdominal cavity. The renal blood vessels and the ureter were ligated, and the whole right kidney was removed. During the sham operations, renal capsules were removed. After the surgeries, the incisions were closed with running sutures, and povidone-iodine was applied on the surface of the skin. As a postoperative medication, *sc.* 0.3 mg/kg nalbuphine hydrochloride (Nalbuphine 10 mg/mL, Teva Pharmaceutical Industries Ltd., Debrecen, Hungary) was administered for four days, twice in the first two postoperative days and once in the third and fourth postoperative days. Enrofloxacin antibiotics (Enroxil 75 mg tablets, Krka, Slovenia; dissolved in tap water in 3.5 mg/L end concentration) were administered in the drinking water for 4 days after both surgeries. No animal died after the operations.

### 4.4. Transthoracic Echocardiography

Cardiac morphology and function were assessed by transthoracic echocardiography at week 8, as previously described [19] (Figure 1). Rats were anesthetized with 2% isoflurane (Forane, AESICA, Queenborough Limited, Queenborough, UK). Then, the chest was shaved, and the animal was placed supine on a heating pad. Two-dimensional, M-mode, Doppler, and tissue Doppler echocardiographic examinations were performed by the criteria of the American Society of Echocardiography with a Vivid IQ ultrasound system (General Electric Medical Systems, New York, NY, USA) using a phased array 5.0- to 11.0 MHz transducer (12S-RS probe, General Electric Medical Systems, New York, NY, USA). Data from 3 consecutive heart cycles were analyzed (EchoPac Dimension v201 software, General Electric Medical Systems, New York, NY, USA) by an experienced investigator in a blinded manner. The mean values of the 3 measurements were calculated and used for statistical evaluation. Systolic and diastolic wall thickness parameters were obtained from the parasternal short-axis view at the level of the papillary muscles (anterior and inferior walls) and the long-axis view at the level of the mitral valve (septal and posterior walls). The LV diameters were measured using M-mode echocardiography from long-axis views between the endocardial borders. The cross-sectional fractional shortening was used as a measure of cardiac contractility (fractional shortening = (LV end-diastolic diameter [LVEDD] − LV end-systolic diameter [LVEDD])/LVEDD × 100). Functional parameters, including LV end-diastolic volume (LVEDV) and LV end-systolic volume (LVESV), were calculated on 4-chamber view images delineating the endocardial borders in diastole and systole. The ejection fraction was calculated using the formula (LVEDV − LVESV)/LVEDV × 100. Stroke volume (SV) and cardiac output (CO) were calculated according to the formulas SV = LVEDV-LVESV and CO = SV × HR, respectively. Diastolic function was assessed using pulse-wave Doppler across the mitral valve from the apical 4-chamber view and tissue Doppler images at the lateral and septal mitral annulus. Early mitral flow (E), septal mitral annulus (E’) velocities, and their ratio (E/E’) indicate diastolic function. Heart rate was calculated using pulse-wave Doppler images while measuring transvalvular flow velocity profiles according to the length of 3 consecutive heart cycles measured between the start points of the E waves.

### 4.5. Blood Pressure Measurement

To measure arterial blood pressure in a separated subgroup of animals PE50 polyethylene catheter (Cole-Parmer, Vernon Hills, IL, USA) was inserted into the left femoral artery under sodium pentobarbital anesthesia (Euthasol, Produlab Pharma b.v., Raamsdonksveer, The Netherlands, 40 mg/kg) (Figure 1). Blood pressure measurements were performed between 09:00 and 14:00 h, with an SEN-02 pressure transducer (MDE Ltd., Budapest, Hungary) connected to an EXP-HG-1 amplifier (MDE Ltd., Budapest, Hungary) and WS-DA data acquisition system (MDE Ltd., Budapest, Hungary). The data were analyzed using the S.P.E.L. advanced haemosys software 3.2 (MDE Ltd., Budapest, Hungary) as described previously [19,33].

### 4.6. Urine and Serum Laboratory Parameters

At week 8, the animals were placed into metabolic cages for 24 h to collect urine and to measure urine creatinine and protein levels. Urine creatinine and protein levels were determined by standard laboratory methods as described previously [20,34] to verify the development of CKD.

Blood was collected from the thoracic aorta at week 9 to measure serum urea (carbamide) and creatinine levels to verify the development of CKD. Serum urea and creatinine levels were quantified by the kinetic UV method using urease and glutamate dehydrogenase enzymes and the Jaffe method, respectively [20,34]. The reagents and the platform analyzers were from Roche Diagnostics (Mannheim, Germany).

Creatinine clearance, an indicator of renal function, was calculated according to the standard formula (urine creatinine concentration [μM] × urine volume for 24 h [mL])/(serum creatinine concentration [μM] × 24 × 60 min) as described previously [20,34] (Figure 1). Urine volume and urine creatinine concentration were measured at week 8, and serum urea and creatinine concentrations were determined at week 9.

### 4.7. Hematoxylin-Eosin and Picrosirius Red and Fast Green Stainings

Formalin-fixed paraffin-embedded subvalvular areas of the left ventricles were cut into 5-μm sections and stained with hematoxylin-eosin (HE) or picrosirius red and fast green (PSFG), as described previously [19,20]. Histological slides were scanned with a Pannoramic Midi II scanner (3DHistech Ltd., Budapest, Hungary). Digital slide processing was performed in SlideViewer version 2.6. On the digital HE images, cardiomyocyte cross-sectional areas were measured to verify the development of LVH at the cellular level.

The Biology Image Analysis Software (BIAS 1.0, Single-Cell Technologies Ltd.) was used to evaluate HE images [19,20]. Image preprocessing was followed by deep learning-based cytoplasm segmentation. User-selected objects were forwarded to the feature extraction module, which is configurable to extract properties from the selected cell components. The transverse CM diameter at the nuclear level and CM perimeter were measured in 100 (consecutive) CMs selected based on longitudinal orientation and mononucleation from a single cut surface (digitalized histological slide) of the LV tissue blocks. The BIAS software also calculated CM cross-sectional areas of the same CMs.

Cardiac fibrosis was assessed on PSFG slides with a program developed in-house, as described previously [11,19]. This program briefly determines the proportion of red pixels in LV sections using two simple color filters. For each red–green–blue (RGB) pixel, the program calculates the color of the pixel in the hue-saturation-luminance color space. The first filter is used to detect red portions of the image. The second filter excludes any white (empty) or light gray (residual dirt on the slide) pixels from further processing using a simple RGB threshold. This way, the program groups each pixel into two sets: pixels considered red and pixels considered green but not red, white, or gray. Red pixels in the first set correspond with connective tissue and fibrosis. Green pixels in the second set correspond with cardiac muscle. Dividing the number of elements in the first set by the number of elements in both sets gives the proportion of the connective tissue compartment of the heart area examined.

### 4.8. mRNA Expression Profiling by RT-qPCR

RT-qPCR was performed with gene-specific primers to monitor mRNA expression, as described previously [33,34]. In the case of LV tissue, RNA was isolated using Qiagen RNeasy Fibrous Tissue Mini Kit (Qiagen, Hilden, Germany) and quantified by NanoDrop One Microvolume UV-Vis spectrophotometer (Thermo Fisher Scientific Inc., Waltham, MA, USA). Then, 100 μg of total RNA was reverse transcribed using iScript™ cDNA Synthesis Kit (BioRad Laboratories Inc., Hercules, CA, USA). The cDNAs were analyzed in technical duplicates in 10-μL reaction volumes. Specific primers (*Col1a1:* collagen type I alpha 1 chain, #qRnoCED0007857; *Col3a1*: collagen type III alpha 1 chain, #qRnoCID0005033) and SsoAdvanced™ Universal SYBR^®^ Green Supermix (BioRad Laboratories Inc., Hercules, CA, USA) were applied according to the manufacturer’s instructions, using a BioRad CFX-96 machine, with the accompanying BioRad CFX Manager BioRad Laboratories Inc., Hercules, CA, USA) software, for cycle threshold value analysis. Relative gene expression was calculated using the relative standard curve method. Ribosomal protein lateral stalk subunit P2 (*Rplp2*, forward primer sequence: agcgccaaagacatcaagaa, reverse primer sequence: tcagctcactgatgaccttgtt) was used as a housekeeping control gene for normalization.

### 4.9. Force Measurements in Isolated Cardiomyocytes

Triton-X-100-permeabilized LV CMs from male and female rats were mounted on a mechanical apparatus to measure the isometric force. Triton-X-100 (Thermo Fisher Scientific, Waltham, MA, USA) removed the membranes of CMs and allowed direct access to the intracellular space. Permeabilized CMs were activated by a solution containing saturating [Ca^2+^] (pCa 4.75, where pCa is −lg[Ca^2+^]), to determine the maximal active force (F_max_). Intracellular solutions contained 7 mM CaEGTA for pCa 4.75 or 7 mM EGTA for pCa 9.0, 37.34 mM KCl, 10 mM N,N-bis (2-hydroxyethyl)-2-aminoethanesulfonic acid (BES), 6.24 mM MgCl_2_, 6.99 mM Na_2_ATP, 15 mM Na_2_CrP at pH = 7.2. Submaximal Ca^2+^—activated force (F_active_) was measured at intermediate [Ca^2+^] (pCa 5.4–7.0) by mixing activating (pCa 4.75) and relaxing (pCa 9.0) solutions to reconstruct pCa-force relationships in isolated CMs. Submaximal forces (F_active_) were normalized to the F_max_ to obtain normalized F_active_ values. The pCa-force relationship was fitted to a modified Hill equation, which allowed for the determination of Ca^2+^ sensitivity of force production (pCa_50_), indicating the Ca^2+^ concentration at which 50% of the maximal force is generated. Ca^2+^-independent passive stiffness (F_passive_) was measured in a relaxing solution after a rapid release-restretch maneuver (lasting 30 ms), which helped assess the passive properties of the CMs. Isometric force measurements were measured at a sarcomere length (SL) of 2.3 µm.

### 4.10. Western Immunoblot

In this procedure, cardiomyocytes were isolated and permeabilized from frozen male and female LV heart tissue samples. The CMs were homogenized in a sample buffer (containing 8 M urea, 2 M thiourea, 3% SDS, 75 mM DTT, 50 mM Tris-HCl (pH 6.8), 10% glycerol, bromophenol blue, 40 µM leupeptin, and 10 µM E-64). The homogenization was performed for 45 min using vortexing. Following this, the samples were centrifuged at 16,000× *g* for 5 min, at 24 °C, to separate the supernatant, from which the protein concentration was determined by a dot-blot technique, adjusting it to 2 mg/mL, using bovine serum albumin (BSA) standards as a reference. To separate large myofilament proteins, such as N2B titin (~3200 kDa) and cardiac myosin-binding protein C (cMyBP-C), agarose-strengthened 2% and 4% SDS-polyacrylamide gels were utilized. The overall phosphorylation levels of titin were assessed using Pro-Q^®^ Diamond phosphoprotein staining following the manufacturer’s protocol, while total protein levels were determined by Coomassie blue staining. For site-specific cardiac troponin I (cTnI) and cMyBP-C phosphorylation analysis via Western immunoblotting, the proteins were separated on 12% polyacrylamide gels and then transferred onto nitrocellulose membranes. The membranes were probed with primary antibodies specific to phosphorylation at PKA-dependent sites [Ser-22/23 (1:1000); Ser-282 (1:500), Abcam, Cambridge, UK and Enzo Life Sciences, Farmingdale, New York, USA] or PKC-dependent sites [Ser-43 (1:500); Thr-144 (1:500), Abcam, Cambridge, UK] followed by peroxidase-conjugated secondary antibodies (1:300, anti-rabbit IgG from Sigma-Aldrich, St. Louis, MI, USA). Detection was performed using enhanced chemiluminescence, and the results were documented with the MF-ChemiBIS 3.2 gel documentation system (DNR Bio-Imaging Systems, Ltd., Jerusalem, Israel). All biochemical analyses were carried out on 4–6 samples per group. The uncropped, full-length Western blot images are presented in Appendix A.

### 4.11. Statistical Analysis

Statistical analysis was performed using GraphPad Prism Software (version 8.01, GraphPad Software Inc.). All values are presented as mean ± SEM. The Sample Size Calculator software was used for assessing the sample size (https://sample-size.net/means-sample-sizeclustered/, last accessed on the 10 October 2023). Specific sample numbers used for measurements are described in the corresponding figure legend. Data points over mean ± 2SD values were excluded from the statistical analysis. The data showed normally distributed sample populations by the Shapiro–Wilk test. In the case of echocardiographic, blood pressure, histology, RT-qPCR, and blood as well as urine measurements, data were compared using a Two-Way Analysis of Variance (ANOVA). Holm–Sidak test was used as a *post hoc test*. Western immunoblot assays were performed in triplicates. Signal intensities of protein bands were quantified by using the ImageJ software version 1.5 (National Institutes of Health, Bethesda, MD, USA) and Magic Plot (Magicplot Systems, Saint Petersburg, Russia) software. Differences were evaluated by a one-way ANOVA, followed by the Bonferroni *post hoc* test or Students’ paired *t*-test, when appropriate. *p* values of <0.05 were considered statistically significant.

## Figures and Tables

**Figure 1 ijms-26-02259-f001:**
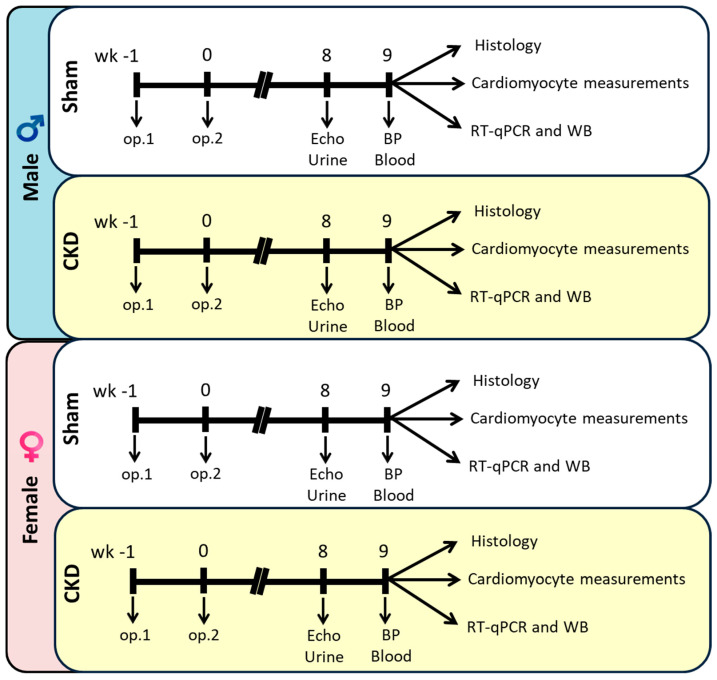
Protocol figure. Male and female Wistar rats underwent a sham operation or a 5/6 nephrectomy in two phases to induce a chronic kidney disease (CKD). In the first stage (op.1), two-thirds of the left kidney was ligated and excised. One week later, the right kidney was removed (op.2). Time-matched sham operations were performed on the control groups. At week 8, cardiac morphology and function were evaluated using transthoracic echocardiography (echo). During this week, the animals were placed in metabolic cages for 24 h to collect urine to measure creatinine and protein levels. In week 9 the rats were anesthetized and blood pressure (BP) was measured. The same week, blood was collected from the thoracic aorta to assess serum urea and creatinine levels. The hearts were then isolated for histology, cardiomyocyte measurements, and biochemical analyses, including quantitative real-time polymerase chain reaction (RT-qPCR) and Western blot (WB).

**Figure 2 ijms-26-02259-f002:**
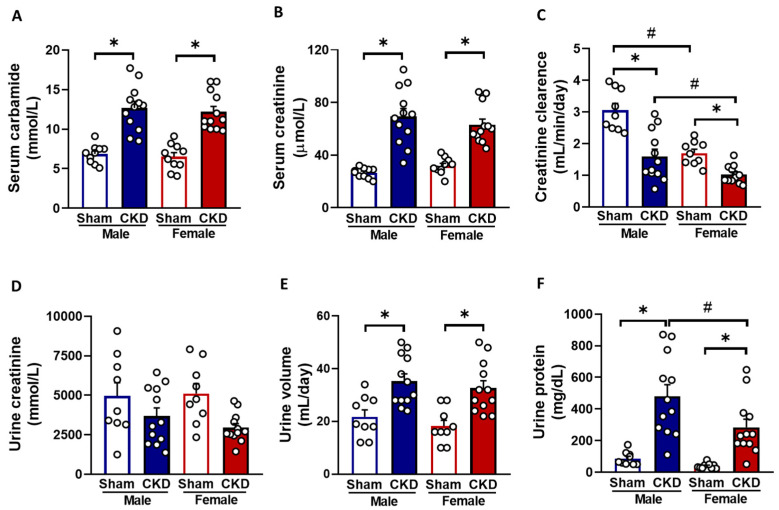
The effects of sex and chronic kidney disease (CKD) on laboratory markers. (**A**) Serum carbamide (urea) concentration, (**B**) serum creatinine concentration, (**C**) creatinine clearance, (**D**) urine creatinine concentration, (**E**) 24 h urine volume, and (**F**) urine protein concentration. Creatinine clearance was calculated using the formula: (urine creatinine concentration [μM] × urine volume for 24 h [mL])/(serum creatinine concentration [μM] × 24 × 60 min). Urine protein and creatinine concentrations and urine volumes were measured at week 8, while serum carbamide and creatinine concentrations were determined at week 9. Values are presented as means  +  SEM, n = 9–12. * *p*  <  0.05, CKD vs. sham-operated groups; # *p*  <  0.05, females vs. males. *p*-values refer to wo-Way ANOVA (Holm–Sidak post hoc test).

**Figure 3 ijms-26-02259-f003:**
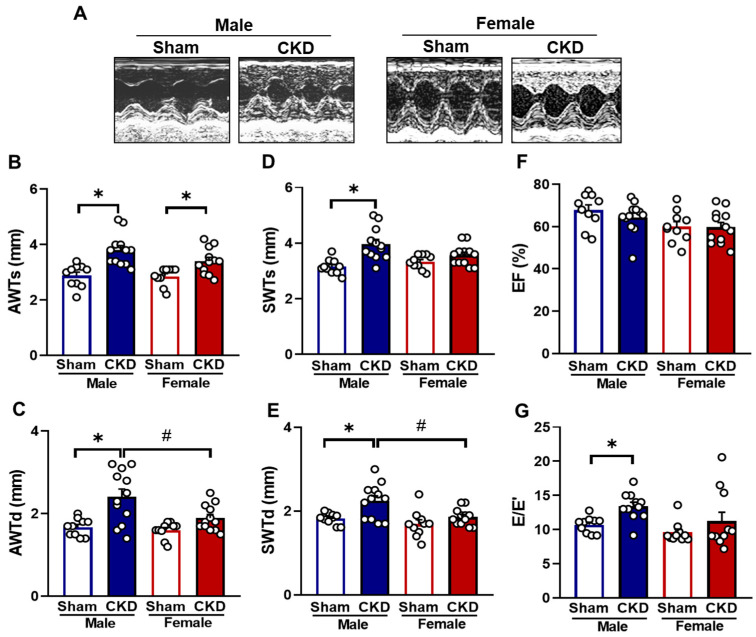
The effects of sex and chronic kidney disease (CKD) on echocardiographic parameters in week 8. (**A**) Representative M-mode images, (**B**) anterior wall thicknesses in systole (AWTs), (**C**) anterior wall thicknesses in diastole (AWTd), (**D**) septal wall thicknesses in systole (SWTs), (**E**) septal wall thicknesses in diastole (SWTd), (**F**) ejection fraction (**E**,**F**), and (**G**) E/E′ ratio. Values are means  +  SEM, n  =  10–12, * *p*  <  0.05, CKD vs. sham-operated groups, # *p*  <  0.05, females vs. males. *p*-values refer to Two-Way ANOVA (Holm–Sidak post hoc test).

**Figure 4 ijms-26-02259-f004:**
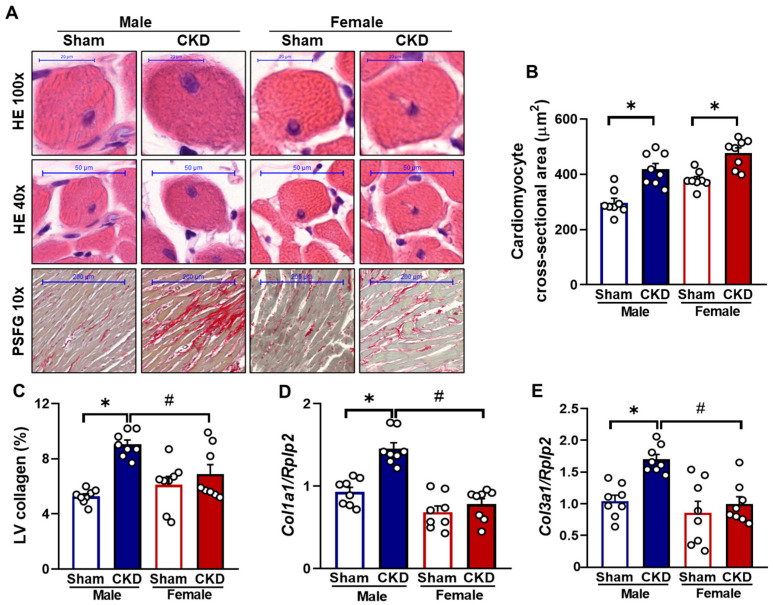
The effects of sex and chronic kidney disease (CKD) on cardiomyocyte hypertrophy and fibrosis at week 9. (**A**) Representative hematoxylin-eosin (HE, 40× and 100×) and picrosirius red/fast green-stained (PSFG, 10×) sections, (**B**) cardiomyocyte cross-sectional area, (**C**) left ventricular (LV) collagen content, (**D**) left ventricular collagen type I alpha 1 (Col1a1) and (**E**) collagen type III alpha 1 (Col3a1) expressions normalized to ribosomal protein lateral stalk subunit P2 (Rplp2) housekeeping gene. Values are presented as means  +  SEM, n  =  8. * *p*  <  0.05, CKD vs. sham-operated groups; # *p*  <  0.05, females vs. males. *p*-values refer to Two-Way ANOVA (Holm–Sidak post hoc test).

**Figure 5 ijms-26-02259-f005:**
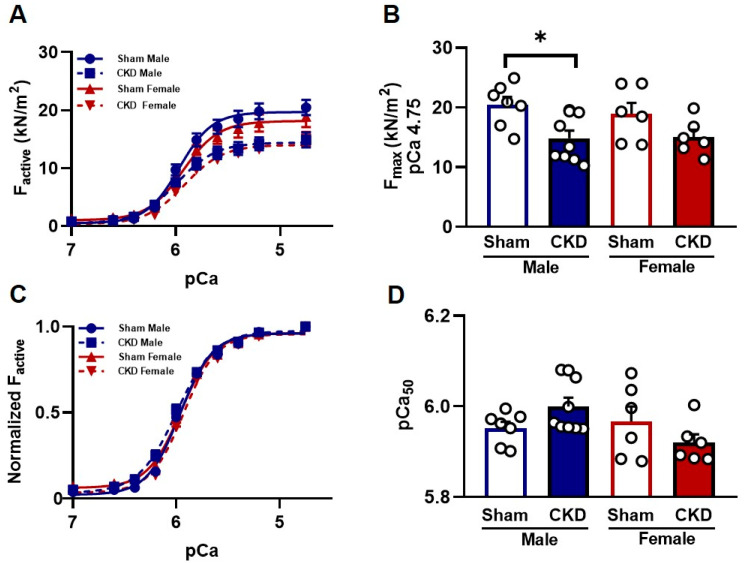
Effects of chronic kidney disease (CKD) on male and female cardiomyocyte active force production. (**A**) Absolute values of active force (F_active_) production in isolated myocyte-sized preparations, at different Ca^2+^ concentrations, in the male and female sham-operated and CKD groups. (**B**) Measurements of maximum (pCa [i.e., −10log[Ca^2+^]] 4.75) Ca^2+^-dependent active (F_max_) force levels in male and female sham-operated and CKD rats (* *p*  <  0.05). (**C**) F_active_ values at submaximal Ca^2+^ concentrations (pCa 5.4–7.0) were expressed relative to F_max_ (pCa 4.75) to determine normalized pCa-force relationships. (**D**) Bar graphs highlight the midpoints of the normalized pCa–force relationships. Calcium sensitivity of force production (pCa_50_) was determined in skinned cardiomyocytes derived from LV tissue in the four animal groups. The sarcomere length was adjusted to 2.3 µm. Data are expressed as mean + SEM, * *p* < 0.05; CKD vs. sham-operated groups, n = 6–9 cardiomyocytes from at least four different hearts.

**Figure 6 ijms-26-02259-f006:**
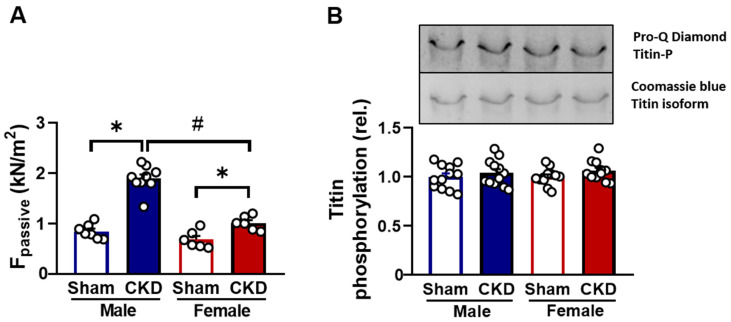
Effects of chronic kidney disease (CKD) on male and female cardiomyocyte passive force production. (**A**) Cardiomyocyte passive tension (F_passive_) and titin protein phosphorylation in the left ventricles in the four experimental groups. F_passive_ was measured at a sarcomere length of 2.3 μm (n = 6–9 cardiomyocytes from at least four different hearts/groups). (**B**) Overall phosphorylation level of the N2B titin isoform was assessed by Pro-Q^®^ Diamond phosphoprotein staining (Thermo Fisher Scientific, Waltham, MA, USA). Titin amounts were visualized by Coomassie-blue staining. Overall titin phosphorylation levels were as follows: male sham-operated and CKD (1.00 ± 0.03 and 1.03 ± 0.04); female sham-operated and CKD (1.00 ± 0.02 and 1.04 ± 0.03 all in relative units). Data are given as mean + SEM, * *p* < 0.05 CKD vs. sham-operated groups; # *p* < 0.05 females vs. males. The uncropped, full-length Western blot images are presented in Appendix A. (See Appendix AA,B for titin phosphorylation).

**Figure 7 ijms-26-02259-f007:**
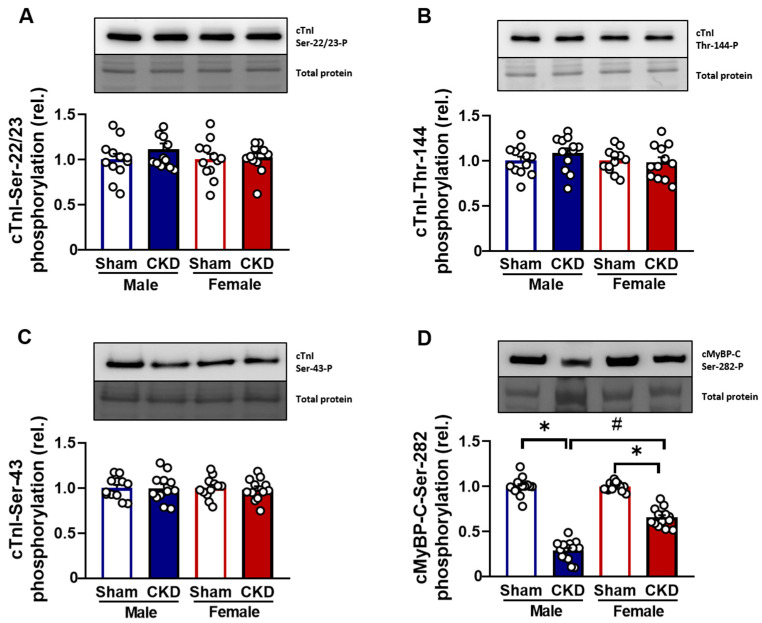
PKA- and PKC-specific phosphorylation of cardiac troponin (cTnI) and cardiac myosin-binding protein C (cMyBP-C) in chronic kidney disease (CKD)-induced uremic cardiomyopathy. (**A**) Phosphorylation levels of cTnI at the Ser-22/23, (**B**) Thr-144, (**C**) Ser-43, and (**D**) cMyBP-C. The phosphorylation levels of the Ser-282 residues were determined by Western immunoblotting in LV cardiomyocytes. The upper bands reflect the phosphorylation status of proteins, and the lower bands indicate total protein amounts. The phosphorylation sites of cTnI and cMyBP-C were labeled with specific antibodies. Total protein amounts were assessed by a super-sensitive blot stain. Bars represent means + SEM (normalized to the mean of the sham-operated group as control), * *p* < 0.05, CKD vs. sham-operated groups, # *p* < 0.05 females vs. males, n = 12–14 independent determinations from at least 4–5 different hearts. The uncropped, full-length Western blot images are presented in Appendix A. (Please see Appendix A for cTnI-Ser22/23, Appendix A for cTnI-Thr144, Appendix A for cTnI-Ser43, and Appendix A for cMyBP-C-Ser-282 phosphorylation).

**Table 1 ijms-26-02259-t001:** The effects of sex and chronic kidney disease (CKD) on LV morphology and function, assessed by echocardiography at week 8. AWTd/s: anterior wall thickness in diastole/systole; CO: cardiac output; EF: ejection fraction; FS: fractional shortening; HR: heart rate; IVRT: isovolumic relaxation time; IWTd/s: inferior wall thickness in diastole/systole; LVEDD and LVESD: left ventricular end-diastolic and end-systolic diameter; LVEDV and LVESV: left ventricular end-diastolic and end-systolic volume; PWTd/s: posterior wall thickness in diastole/systole; SV: stroke volume; SWTd/s: septal wall thickness in diastole/systole. Values are means  ±  SEM, n  =  10–12, * *p*  <  0.05, CKD vs. sham-operated groups, # *p*  <  0.05, females vs. males. *p*-values refer to Two-Way ANOVA (Holm–Sidak post hoc test).

Parameter (Unit)	Male		Female	
	Sham	CKD	Sham	CKD
AWTs (mm)	2.88 ± 0.13	3.80 ± 0.17 *	2.86 ± 0.10	3.40 ± 0.14 *
AWTd (mm)	1.67 ± 0.07	2.41 ± 0.19 *	1.60 ± 0.06	1.89 ± 0.10 #
IWTs (mm)	3.10 ± 0.14	3.45 ± 0.16	3.35 ± 0.12	3.64 ± 0.13
IWTd (mm)	1.85 ± 0.11	2.00 ± 0.10	1.76 ± 0.12	1.98 ± 0.11
SWTs (mm)	3.15 ± 0.09	3.97 ± 0.17 *	3.33 ± 0.08	3.57 ± 0.11
SWTd (mm)	1.82 ± 0.04	2.42 ± 0.12 *	1.69 ± 0.11	1.86 ± 0.06 #
PWTs (mm)	2.98 ± 0.13	3.43 ± 0.14 *	3.29 ± 0.14	3.45 ± 0.18
PWTd (mm)	1.76 ± 0.10	1.94 ± 0.11	1.86 ± 0.08	1.92 ± 0.14
FS (%)	46.01 ± 1.31	56.90 ± 3.90 *	64.90 ± 3.20 #	67.71 ± 2.30 #
LVEDD (mm)	7.57 ± 0.36	7.15 ± 0.29	6.22 ± 0.24 #	6.06 ± 0.10 #
LVESD (mm)	4.07 ± 0.20	3.16 ± 0.32 *	2.22 ± 0.28 #	1.97 ± 0.15 #
LVEDV (μL)	106.70 ± 6.50	69.51 ± 3.91 *	82.80 ± 6.61 #	95.30 ± 6.30 #
LVESV (μL)	34.81 ± 4.61	24.01 ± 2.20	33.20 ± 3.60	38.51 ± 4.11 #
SV (μL)	71.90 ± 4.20	45.52 ± 3.61 *	49.70 ± 4.31 #	56.81 ± 4.11 #
HR (1/min)	352 ± 10	350 ± 10	368 ± 14	377 ± 11 #
CO (mL/min)	24.41 ± 1.61	15.52 ± 1.42 *	18.61 ± 2.02 #	22.12 ± 1.71
EF (%)	67.81 ± 2.52	65.75 ± 2.52	60.01 ± 2.71	59.62 ± 2.52
E-velocity (m/s)	0.86 ± 0.05	0.80 ± 0.04	0.86 ± 0.06	0.90 ± 0.05
e’-velocity (m/s)	0.08 ± 0.01	0.06 ± 0.01	0.09 ± 0.01	0.08 ± 0.01
E/e’	10.71 ± 0.41	13.41 ± 0.62 *	9.62 ± 0.63 #	11.21 ± 1.34 #
IVRT (ms)	23.92 ± 1.11	29.01 ± 1.51 *	23.13 ± 1.92	23.62 ± 1.53 #

**Table 2 ijms-26-02259-t002:** The effects of sex and chronic kidney disease (CKD) on blood pressure and ex vivo morphometric parameters at week 9. Values are means  ±  SEM, n = 7–11 for blood pressure and n  =  10–12 for ex vivo data, * *p*  <  0.05, CKD vs. sham-operated groups, # *p*  <  0.05, females vs. males. *p*-values refer to Two-Way ANOVA (Holm–Sidak post hoc test).

Parameter (Unit)	Male		Female	
	Sham	CKD	Sham	CKD
Systolic blood pressure (mmHg)	127 ± 6	120 ± 4	123 ± 6	131 ± 7
Diastolic blood pressure (mmHg)	100 ± 6	94 ± 4	95 ± 7	103 ± 8
Mean blood pressure (mmHg)	111 ± 6	105 ± 4	107 ± 7	114 ± 8
Body weight	508 ± 20	519 ± 9	327 ± 10 #	324 ± 5 #
Tibia length (cm)	4.29 ± 0.04	4.3 ± 0.05	3.92 ± 0.04 #	3.93 ± 0.03 #
Heart weight (mg)	1158 ± 35	1324 ± 47 *	946 ± 34 #	1113 ± 56 *#
Kidney weight (mg)	1330 ± 43	1809 ± 79 *	989 ± 50 #	1189 ± 75 *#
Lung weight (mg)	1596 ± 55	1765 ± 67 *	1508 ± 36 #	1464 ± 37 #

## Data Availability

The data generated and analyzed during the current study are available from the corresponding author on a reasonable request.

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
