# Peer review of "Differential Myocardial Responses in Male and Female Rats with Uremic Cardiomyopathy"

_ijms, 2025, doi:10.3390/ijms26052259_

Round 1

Reviewer 1 Report

Comments and Suggestions for Authors

The article is quite interesting, and I have only minor suggestions for the authors.

From my perspective, the authors should discuss the possibility of adaptability regarding residual kidney volume, including enlargement and adaptation. While the authors demonstrate an increase in creatinine levels in the operated animals, there is also a period of time between the two interventions during which the intact kidney could compensate for the loss of some function following the first operation, and how this time could impact the results.

The authors could provide data on any differences in volume and dimensions of the right kidney between the two surgeries.

Reviewer 2 Report

Comments and Suggestions for Authors

The authors presented an interesting article regarding myocardial responses in rats with uremic cardiomyopathy with a focus on sex-specific differences. Indeed, the quality of the manuscript and the study is at a high level, and I would recommend publication after minor adjustments:

Results:

Table 1: Please check the font. Important statistical findings could be bolded for better visualization. The same is for Table 2.

The Discussion:

The authors could add the limitations of their study.

Reviewer 3 Report

Comments and Suggestions for Authors

The manuscript by Bodi and colleagues provides insight into sex differences in cardiovascular complications arising from chronic kidney disease (CKD). While confirming previous findings that both sexes exhibit similar susceptibility to cardiomyocyte hypertrophy following CKD, this study extends the analysis by demonstrating a more severe myocardial remodeling in males. Specifically, although serum markers confirm kidney dysfunction in both sexes after subtotal nephrectomy, male model rats display more pronounced signs of uremic cardiomyopathy and left ventricular hypertrophy than females. Moreover, histological analysis reveals severe fibrosis exclusively in male animals.

The study's results reinforce previous evidence linking increased left ventricular passive stiffness to changes in sarcomeric myofilament proteins in both sexes, particularly cMyBP-C hypophosphorylation. This alteration impairs diastolic relaxation, exacerbating heart failure symptoms.

Despite its limited novelty, this study provides a significant contribution by elucidating sex-based differences in uremic cardiomyopathy. The varying response to pathological stimuli and therapeutic interventions remains a poorly explored area in medical research, making these findings highly relevant.

The experiments are well conducted and the limitations of the study are clearly underlined. Regrettably, the present work lacks a mechanistic explanation for the observed sex-based protection against severe cardiomyopathy in females.
